# Towards Training GNNs using Explanation Directed Message Passing

**Valentina Giunchiglia**[*]
Imperial College London
v.giunchiglia20@imperial.ac.uk

**Chirag Varun Shukla**[*]
Ludwig-Maximilians Universität München
shukla@math.lmu.de

**Guadalupe Gonzalez**
Imperial College London
ggg17@ic.ac.uk

**Chirag Agarwal**
Adobe
chiragagarwall12@gmail.com

## Abstract

With the increasing use of Graph Neural Networks (GNNs) in critical real-world applications, several post hoc explanation methods have been proposed to understand their predictions. However, there has been no work in generating explanations on the fly during model training and utilizing them to improve the expressive power of the underlying GNN models. In this work, we introduce a novel explanation-directed neural message passing framework for GNNs, EXPASS (EXplainable message PASSing), which aggregates only embeddings from nodes and edges identified as important by a GNN explanation method. EXPASS can be used with any existing GNN architecture and subgraph-optimizing explainer to learn accurate graph embeddings. We theoretically show that EXPASS alleviates the oversmoothing problem in GNNs by slowing the layer-wise loss of Dirichlet energy and that the embedding difference between the vanilla message passing and EXPASS framework can be upper bounded by the difference of their respective model weights. Our empirical results show that graph embeddings learned using EXPASS improve the predictive performance and alleviate the oversmoothing problems of GNNs, opening up new frontiers in graph machine learning to develop explanation-based training frameworks.

## 1 Introduction

Graph Neural Networks (GNNs) are increasingly used as powerful tools for representing graph-structured data, such as social, information, chemical, and biological networks [1, 2]. With the deployment of GNN models in critical applications (e.g., financial systems and crime forecasting [3, 4]), it becomes essential to ensure that the relevant stakeholders understand and trust their decisions. To this end, several approaches [5–13] have been proposed in recent literature to generate *post hoc* explanations for predictions of GNN models.

In contrast to other modalities like images and texts, generating instance-level explanations for graphs is non-trivial. In particular, it is more challenging since individual node embeddings in GNNs aggregate information using the entire graph structure, and, therefore, explanations can be on different levels (i.e., node attributes, nodes, and edges). While several categories of GNN explanation methods have been proposed: gradient-based [5, 10, 14], perturbation-based [8, 9, 11, 13, 15], and surrogate-based [7, 12], their utility is limited to generating post hoc node- and edge-level explanations for a given pre-trained GNN model. Thus, the capability of GNN explainers to improve the predictive performance of a GNN model lacks understanding as there is very little work on systematically analyzing the reliability of state-of-the-art GNN explanation methods on model performance [16].

---

[*]Equal contribution.

Giunchiglia et al., Towards Training GNNs using Explanation Directed Message Passing. *Proceedings of the First Learning on Graphs Conference (LoG 2022)*, PMLR 198, Virtual Event, December 9–12, 2022.

To address this, recent works have explored the joint optimization of machine learning models and explanation methods to improve the reliability of explanations [17, 18]. Zhou et al. [18] proposed DropEdge as a technique to drop random edges (similar to generating random edge explanations) during training to reduce overfitting in GNNs. More recently, Spinelli et al. [17] used meta-learning frameworks to generate GNN explanations and show an improvement in the performance of specific GNN explanation methods. While these works make an initial attempt at jointly optimizing explainers and predictive models, they are neither generalizable nor exhaustive. They fail to show improvement in the downstream GNN performance [17] and degree of explainability [18] across diverse GNN architectures and explainers. Further, there is little to no work done on either theoretically analyzing the effect of GNN explanations on the neural message framework in GNNs or on important GNN properties like oversmoothing [19].

**Present work.** In this work, we introduce a novel explanation-directed neural message passing framework, EXPASS, which can be used with any GNN model and subgraph-optimizing explainer to learn accurate graph representations. In particular, EXPASS utilizes GNN explanations to steer the underlying GNN model to learn graph embeddings using only important nodes and edges. EXPASS aims to define local neighborhoods for neural message passing, i.e., identify the most important edges and nodes, using explanation weights, in the $k$-hop local neighborhood of every node in the graph. Formally, we augment existing message passing architectures to allow information flow along important edges while blocking information along irrelevant edges.

We present an extensive theoretical and empirical analysis to show the effectiveness of EXPASS on the predictive, explainability, and oversmoothing performance of GNNs. Our theoretical results show that the embedding difference between vanilla message passing and EXPASS frameworks is upper-bounded by the difference between their model weights. Further, we show that embeddings learned using EXPASS relieve the oversmoothing problem in GNNs as they reduce information propagation by slowing the layer-wise loss of Dirichlet energy (Section 4.2). For our empirical analysis, we integrate EXPASS into state-of-the-art GNN models and evaluate their predictive, oversmoothing, and explainability performance on real-world graph datasets (Section 5). Our results show that, on average, across five GNN models, EXPASS improves the degree of explainability of the underlying GNNs by 39.68%. Our ablation studies show that for an increasing number of GNN layers, EXPASS achieves 34.4% better oversmoothing performance than its vanilla counterpart. Finally, our results demonstrate the effectiveness of using explanations during training, paving the way for new frontiers in GraphXAI research to develop explanation-based training algorithms.

## 2   Related works

**Graph Neural Networks.** Graph Neural Networks (GNNs) are complex non-linear functions that transform input graph structures into a lower dimensional embedding space. The main goal of GNNs is to learn embeddings that reflect the underlying input graph structure, i.e., neighboring nodes in the graph are mapped to neighboring points in the embedding space. Prior works have proposed several GNN models using spectral and non-spectral approaches. Spectral models [20–24] leverage Fourier transform and graph Laplacian to define convolution approaches for GNN models. However, non-spectral approaches [25–29] define the convolution operation by leveraging the local neighborhood of individual nodes in the graph. Most modern non-spectral models are message-passing frameworks [30, 31], where nodes update their embedding by aggregating information from $k$-hop neighboring nodes.

**Post hoc Explanations.** With the increasing development of complex high-performing GNN models [25–29], it becomes critical to understand their decisions. Prior works have focused on developing several post hoc explanation methods to explain the decisions of GNN models [5, 7, 9, 11–13, 32]. More specifically, these explanation methods can be broadly categorized into i) gradient-based methods [5] that leverage the gradients of the GNN model to generate explanations; ii) perturbation-based methods [9, 11, 13] that aim to generate explanations by calculating the change in GNN predictions upon perturbations of the input graph structure (nodes, edges, or subgraphs); and iii) surrogate-based methods [7, 12] that fit a simple interpretable model to approximate the predictive behavior of the given GNN model. Finally, recent works have introduced frameworks to theoretically and empirically analyze the behavior of state-of-the-art GNN explanation methods with respect to several desirable properties [16, 33].

# 3 Preliminaries

**Notations.** Let $\mathcal{G} = (\mathcal{V}, \mathcal{E}, \mathbf{X})$ denote an undirected graph comprising of a set of nodes $\mathcal{V}$ and a set of edges $\mathcal{E}$. Let $\mathbf{X} = \{\mathbf{x}_1, \mathbf{x}_2, \ldots, \mathbf{x}_N\}$ denote the set of node feature vectors for all nodes in $\mathcal{V}$, where $\mathbf{x}_v \in \mathbb{R}^d$ captures the attribute values of a node $v$ and $N = |\mathcal{V}|$ denotes the number of nodes in the graph. Let $\mathbf{A} \in \mathbb{R}^{N \times N}$ be the graph adjacency matrix, where element $\mathbf{A}_{uv} = 1$ if there exists an edge $e \in \mathcal{E}$ between nodes $u$ and $v$ and $\mathbf{A}_{uv} = 0$ otherwise. We use $\mathcal{N}_u$ to denote the set of immediate neighbors of node $u$, *i.e.,* , $\mathcal{N}_u = \{v \in \mathcal{V} | A_{uv} = 1\}$. Finally, the function deg $: \mathcal{V} \mapsto \mathbb{Z}_{>0}$ is defined as $\deg(v) = |\mathcal{N}_v|$ and outputs the degree of a node $v \in \mathcal{V}$

**Graph Neural Networks (GNNs).** Formally, GNNs can be formulated as message passing networks [30] specified by three key operators MSG, AGG, and UPD. These operators are recursively applied on a given graph $\mathcal{G}$ for a $L$-layer GNN model defining how neural messages are shared, aggregated, and updated between nodes to learn the final node representations in the $L^{\text{th}}$ layer of the GNN. Commonly, a message between a pair of nodes $(u, v)$ in layer $l$ is characterized as a function of their hidden representations $\mathbf{h}_u^{(l-1)}$ and $\mathbf{h}_v^{(l-1)}$ from the previous layer: $\mathbf{m}_{uv}^{(l)} = \text{MSG}(\mathbf{h}_u^{(l-1)}, \mathbf{h}_v^{(l-1)})$. The AGG operator retrieves the messages from the neighborhood of node $u$ and aggregates them as: $\mathbf{m}_u^{(l)} = \text{AGG}(\mathbf{m}_{uv}^{(l)} \mid v \in \mathcal{N}_u)$. Next, the UPD operator takes the aggregated message $\mathbf{m}_u^{(l)}$ at layer $l$ and combines it with $\mathbf{h}_u^{(l-1)}$ to produce node $u$'s representation for layer $l$ as $\mathbf{h}_u^{(l)} = \text{UPD}(\mathbf{m}_u^{(l)}, \mathbf{h}_u^{(l-1)})$. Lastly, the final node representation for node $u$ is given as $\mathbf{z}_u = \mathbf{h}_u^{(L)}$.

**Graph Explanations.** In contrast to other modalities like images and texts, an explanation method for graphs can formally generate multi-level explanations. For instance, in a graph classification task, the explanations for a given graph prediction can be with respect to the node attributes $\mathbf{M}_x \in \mathbb{R}^d$, nodes $\mathbf{M}_n \in \mathbb{R}^N$, or edges $\mathbf{M}_e \in \mathbb{R}^{N \times N}$. Note that these explanation masks are continuous but can be discretized using specific thresholding strategies [33].

**Oversmoothing.** Cai et al. [34] and Zhou et al. [35] defined bounds for analyzing oversmoothing for a GNN using Dirichlet Energy. For a graph $\mathcal{G}$ with adjacency matrix $\mathbf{A}$ and degree matrix $\mathbf{D}$, we define $\tilde{\mathbf{A}} = \mathbf{A} + \mathbf{I}_N$ and $\tilde{\mathbf{D}} = \mathbf{D} + \mathbf{I}_N$ as the adjacency and degree matrices respectively of the graph $\mathcal{G}$ with self-loops. We also define the augmented normalized Laplacian of $\mathcal{G}$ as $\tilde{\Delta} = \mathbf{I}_N - \tilde{\mathbf{D}}^{-\frac{1}{2}} \tilde{A} \tilde{D}^{-\frac{1}{2}}$, and $\mathbf{P} = \mathbf{I}_N - \tilde{\Delta}$.

# 4 Our Framework: EXPASS

Here, we describe EXPASS, our proposed explainable message-passing framework that aims to learn accurate and interpretable graph embeddings. In particular, EXPASS incorporates explanations into the message-passing framework of GNN models by only aggregating embeddings from key nodes and edges as identified using an explanation method.

**Problem formulation (Explanation Directed Message Passing).** *Given a graph $\mathcal{G} = (\mathcal{V}, \mathcal{E}, \mathbf{X})$, EXPASS aims to generate a $d$-dimensional embedding $\mathbf{z}_u \in \mathbb{R}^d$ for each node $u \in \mathcal{V}$ using an explanation-directed message passing framework that filters out the noise from unimportant edges and improves the expressive power of GNNs.*

## 4.1 Explanation Directed Message Passing

The central idea of EXPASS is to propose a novel method for improving the neural message passing scheme of GNN models by utilizing explanations during model training and aggregating important neural messages along edges in graph neighborhoods. Next, we describe the existing message-passing scheme in GNNs and our explainable counterpart.

**Message Passing.** As described in Section 3, each GNN layer can be described using the MSG, AGG, and UPD operators. For each node $u \in \mathcal{V}$, the $(l+1)^{th}$ layer embeddings $\mathbf{h}_u^{(l+1)}$ is computed using a GNN operating on the node's neighboring attributes. Formally, the GNN layer can be formulated as:

$$\mathbf{h}_u^{(l+1)} = \phi\left(\mathbf{h}_u^{(l)}, \bigoplus_{v \in \mathcal{N}_u} \psi(\mathbf{h}_u^{(l)}, \mathbf{h}_v^{(l)})\right)$$

UPDATE AGGREGATE MESSAGE

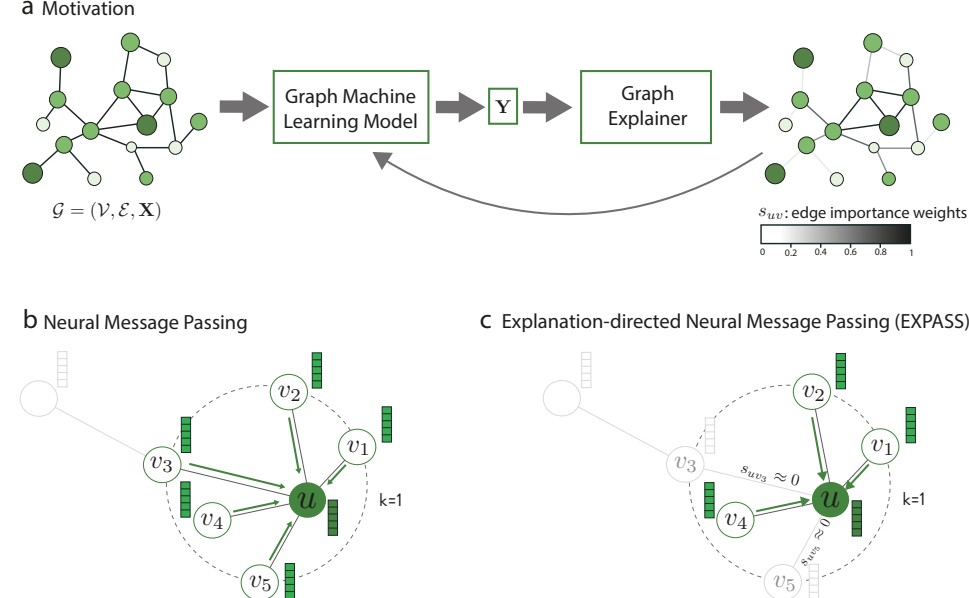

Figure 1: **Overview of EXPASS: a)** EXPASS investigates the problem of injecting explanations into the message-passing framework to increase the expressive power and performance of GNNs. **b)** Shown is the general message passing scheme where, for node $u$, messages are aggregated from nodes $v_i \in \mathcal{N}_u$ in the 1-hop neighborhood of $u$. **c)** EXPASS injects explanations into the message passing framework translates by masking out messages from neighboring nodes $v_i \in \mathcal{N}_u$ with explanation scores $s_{uv_i} \approx 0$ when $u$ is correctly classified.

where $\mathbf{h}_u^{(l+1)}$ represents the updated embedding of node $u$, $\psi$ is the MSG operator, $\bigoplus$ is the AGG operator (*e.g.,* summation), $\phi$ is an UPD function (*e.g.,* any non-linear activation function), and $\mathbf{h}_u^{(l)}$ represents the embedding of node $u$ from the previous layer. We obtain an embedding $\mathbf{z}_u$ for node $u$ by stacking $L$ GNN layers. Finally, the node embeddings $\mathbf{Z} \in \mathbb{R}$ are then passed to a READOUT function to obtain an embedding for the graph.

**EXPASS.** Here, we describe our proposed explainable message-passing scheme that incorporates explanations into the message-passing step in individual GNN layers on the fly during the training process. Given an explanation method, which generates an importance score $s_{uv} \in \mathbf{M}_u^e$ for every edge $e_{uv} \in \mathcal{E}$, we can weight the edge contribution in the neighborhood $\mathcal{N}_u$ of node $u$ as:

$$\mathbf{h}_u^{'(l+1)} = \phi\left(\mathbf{h}_u^{(l)}, \bigoplus_{v \in \mathcal{N}_u} \boxed{s_{uv}}\, \psi(\mathbf{h}_u^{(l)}, \mathbf{h}_v^{(l)})\right)$$

Note that EXPASS is agnostic to explanation types and can also incorporate explanations on node attributes and node level. For instance, the importance scores for individual nodes can be computed by averaging the outgoing scores $s_{uv}$ for all $v \in \mathcal{N}_u$. Subsequently, we can replace the $s_{uv}$ score by using the average score $s_u$ to weight edges in the EXPASS layers, and for node attributes, we can multiply the node attribute explanation $\mathbf{M}_u^a$ to the original node attribute vector.

To enable explainable message passing and only retain the important embeddings for node $u$, EXPASS removes the knowledge of irrelevant nodes and edges from the local neighborhood $\mathcal{N}_u$ of node $u$ using its explanations. For instance, if node $v$ is considered important to node $u$, EXPASS transforms the aggregated messages of node $u$ using the node importance scores $s_{uv}$. Note that since the explanations of node $u$ include important nodes and edges in the $L$-hop neighborhood of node $u$, even though node $u$ is only locally modified, the change will spread through all the nodes in every GNN layer. Furthermore, to avoid spurious correlations, we ensure that explanations are only generated for correctly classified nodes and graphs. Explanation weights infuse information from higher-order neighborhoods into each layer of the GNN model, specifically, from as many $L$-hop neighbors

because explanation weights within each layer are computed using the $L$-layer GNN model. To illustrate this, we next show the weight computations for a GNN explanation method.

Without loss of generality, let us consider GNNExplainer as our explanation method whose mask for the selected graph is formulated as: $\mathcal{G}_{\text{mask}} = (\mathbf{X}', \mathbf{A}') = (\mathbf{X} \odot \sigma(\mathbf{M}^{\text{x}}), \mathbf{A} \odot \sigma(\mathbf{M}^{\text{e}}))$, where $W = [\mathbf{M}^{\text{x}}, \mathbf{M}^{\text{e}}]$ are the explainers parameters, $\sigma$ is the sigmoid function, and $\odot$ denotes element-wise multiplication. Here, $s_{uv}$ represents the element in row $v$ and column $u$ of $\mathbf{M}^{\text{e}}$. Gradient descent-based optimization is used to find the optimal values for the masks minimizing the following objective: $L_e = -\sum_{c=1}^{C} 1[y = c] \log f_\theta(Y = y | \mathcal{G}_{\text{mask}})$, where $f_\theta$ is the $L$-layer GNN model and $C$ is the total number of classes. This shows that a $L$-hop neighborhood is used to compute $s_{uv}$. Formally, it minimizes the uncertainty of the predictive model when the GNN computation is limited to the explanation subgraph. This uncertainty is minimized as a proxy of the maximization of the mutual information between the prediction with the unmasked graph and masked graph.

## 4.2 Theoretical Analysis

Here, we provide a detailed theoretical analysis of our proposed EXPASS framework. In particular, we (i) provide a theoretical upper bound on the embedding difference obtained from a vanilla message passing and EXPASS framework and (ii) show that graph embeddings learned using EXPASS relieves the oversmoothing problem in GNNs by reducing information propagation.

**Theorem 1** (Differences between EXPASS and Vanilla Message Passing). *Given a non-linear activation function $\sigma$ that is Lipschitz continuous, the difference between the node embeddings between a vanilla message passing and* EXPASS *framework can be bounded by the difference in their individual weights, i.e.,*

$$\|\mathbf{h}_u^{(l)} - \mathbf{h}'^{(l)}_u\|_2 \leq \|\mathbf{W}_a^{(l)} - \mathbf{W}'^{(l)}_a\|_2 \|\mathbf{h}_u^{(l-1)}\|_2 + \|\mathbf{W}_n^{(l)} - \mathbf{W}'^{(l)}_n\|_2 \sum_{v \in \mathcal{N}_u \cap s_v = 1} \|\mathbf{h}_v^{(l-1)}\|_2, \quad (1)$$

*where $\mathbf{W}_a^{(l)}$ and $\mathbf{W}'^{(l)}_a$ are the weights for node $u$ in layer $l$ of the vanilla message passing and* EXPASS *framework and $\mathbf{W}_n^{(l)}$ and $\mathbf{W}'^{(l)}_n$ are their weight matrix with neighbors of node $u$ at layer $l$.*

*Proof Sketch.* In Theorem 1, we prove that the $\ell_2$-norm of the differences between the embeddings of vanilla message passing and EXPASS framework at layer $l$ is upper bounded by the difference between their weights and the embeddings of node $u$ and its subgraph. See Appendix A for more details. $\square$

**Definition 1** (Dirichlet Energy for a Node Embedding Matrix [35]). *Given a node embedding matrix $\mathbf{H}^{(l)} = [\mathbf{h}_1^{(l)}, \ldots, \mathbf{h}_n^{(l)}]^T$ learned from the GNN model at the $l^{th}$ layer, the Dirichlet Energy $E(\mathbf{H}^{(l)})$ is defined as:*

$$E(\mathbf{H}^{(l)}) = tr(\mathbf{H}^{(l)^T} \tilde{\Delta} \mathbf{H}^{(l)}) = \frac{1}{2} \sum_{i,j \in \mathcal{V}} a_{ij} \|\frac{\mathbf{H}_i^{(l)}}{\sqrt{1 + deg_i}} - \frac{\mathbf{H}_j^{(l)}}{\sqrt{1 + deg_j}}\|_2^2 \quad (2)$$

*where $a_{ij}$ are elements in the adjacency matrix $\tilde{\mathbf{A}}$ and $deg_i, deg_j$ is the degree of node $i$ and $j$, respectively.*

Cai et al. [34] extensively show that higher Dirichlet energies correspond to lower oversmoothing. Furthermore, they show that the removal of edges or, similarly, the reduction of edge weights on graphs helps alleviate oversmoothing.

**Proposition 1** (EXPASS relieves Oversmoothing). EXPASS *alleviates oversmoothing by slowing the layer-wise loss of Dirichlet energy.*

The complete proof is provided in Appendix A.

## 5 Experiments

Next, we present experimental results for our EXPASS framework. More specifically, we address the following questions: **Q1)** Does EXPASS enable GNNs to learn more accurate embeddings and improve their degree of explainability? **Q2)** How does EXPASS affect the oversmoothing and

predictive performance of GNNs with an increasing number of layers? **Q3)** Does EXPASS depend on the quality of explanations for improving the predictive and oversmoothing performance of GNNs and are they better than attention weights? **Q4)** How does EXPASS helps in the evolution of explanation during the training of the GNN model? [2]

## 5.1  Datasets and Experimental setup

We first describe the datasets used to study the utility of our proposed EXPASS framework and then outline the experimental setup.

**Datasets.** We use real-world molecular chemistry datasets to evaluate the effectiveness of EXPASS w.r.t. the performance of the underlying GNN model and understand the trade-off between explainability and accuracy for a graph classification task. We consider four benchmark datasets, which includes Mutag [36], Alkane-Carbonyl [37], DD [38], and Proteins [39]. See Appendix B.1 for a detailed overview of the datasets.

**GNN Architectures and Explainers.** To investigate the flexibility of EXPASS, we incorporate it into five different GNN models: GCN [40], GraphConv [41], LEConv [42], GraphSAGE [28], GAT [43], and GIN [27]. We use GNNExplainer [13] as our baseline GNN explanation method to generate edge-level explanations for most of our experiments. In addition, we use Integrated Gradients [44] and PGMExplainer [12], a node-level explanation method, to demonstrate EXPASS's sensitivity to the choice of explainers.

**Implementation details.** We consider DropEdge [45] as our baseline method for comparing the oversmoothing performance of EXPASS as DropEdge randomly removes edges from the input graph at each training epoch, acting like a message passing reducer. Across all experiments, we use topK (k=40%) node features/edges, and use them to generate explanations for all explanation methods. All other hyperparameters of the explanation and baseline methods were set following the author's guidelines. For all our experiments (unless mentioned otherwise), we use the baseline architectures with three GNN layers followed by ReLU layers and set the hidden dimensionality to 32. Finally, we use a single linear layer to transform the graph embeddings to their respective classes. See Appendix B.2 for more details.

**Performance metrics for GNN Explainers.** To measure the reliability of GNN explanation methods, we use the graph explanation faithfulness metric [16]: $\text{GEF}(\hat{y}_u, \hat{y}_{u'}) = 1 - \exp^{-\text{KL}(\hat{y}_u || \hat{y}_{u'})}$, where $\hat{y}_u$ is predicted probability vector using the whole subgraph and $\hat{y}_{u'}$ is the predicted probability vector using the masked subgraph, where we generate the masked subgraph by only using the topK features identified by an explanation and the Kullback-Leibler (KL) divergence score (denoted by "$||$" operator) quantifies the distance between two probability distributions. Note that GEF is a measure of the unfaithfulness of the explanation. So, higher values indicate a higher degree of unfaithfulness.

**Performance metrics for Oversmoothing.** Zhou et al. [18] introduced the Group Distance Ratio (GDR) metric to quantify oversmoothing in GNNs. It measures the ratio between the average of pairwise representation distances between graphs belonging to different (inter) and same (intra) groups. Formally, one would prefer to reduce the intra-group class representations and increase the inter-group distance to relieve the over-smoothing issue. Hence, lower GDR values denote higher oversmoothing in GNNs.

**Burn-in period.** We defined the *burn-in period* as a number *n* of epochs during training in which no explanations are used. The burn-in period is necessary to avoid feeding spurious explanations to the model. The length of the burn-in period (i.e., the number of epochs) was treated as a hyperparameter and fine-tuned using the validation set. At the end of the burn-in period, a predefined percentage of correctly predicted graphs per batch is randomly sampled and their explanations are used in the model training. The percentage of correctly predicted graphs sampled in each batch was set to 0.4 for all our experiments. See Appendix C.2 for ablation on burn-in periods.

## 5.2  Results

**Q1) EXPASS improves the predictive performance and explainability of GNNs.** To measure the predictive performance and degree of explainability of GNNs trained using EXPASS, we compute

---

[2]Code to reproduce the results is available here

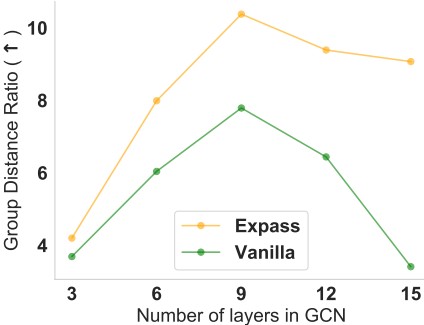 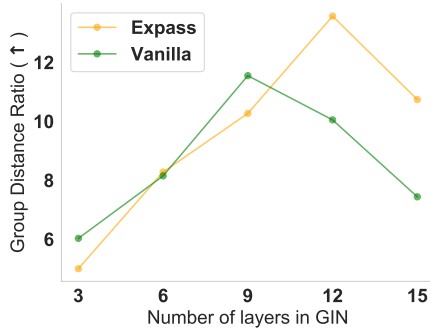

Figure 2: The effects of the number of GNN layers on the oversmoothing performance of EXPASS (orange) and Vanilla (green) GCN (left column) and GIN (right column) models trained on Alkane-Carbonyl dataset. Across models with increasing number of layers, EXPASS achieves higher GDR performance without sacrificing the predictive performance of the GCN model. See Figs. 5-7 for predictive performance results.

their average predictive performance (using AUROC and F1-score) and fidelity (using Graph Explanation Faithfulness) using different GNN models and datasets. Across four datasets and five GNN architectures, we find that EXPASS-augmented GNNs learn graph embeddings that are more accurate (higher AUROC and F1-score) and result in more faithful explanations (lower Graph Explanation Faithfulness score) than their vanilla counterparts. On average, EXPASS improves the AUROC and F1-score by 1.51% and 1.05%, respectively. In particular, we observe that EXPASS improves the predictive behavior of high-performing models like GIN (+2.06% in AUROC and +2.50% in F1-score) but shows little to no improvement in the case of LeConv, which utilizes a node-scoring mechanism through the similarity between a node and its neighbors' embeddings. Finally, we find that EXPASS-augmented GNNs significantly improve the explainability of a GNN and achieve a 39.68% better faithfulness score as compared to vanilla GNNs (Table 1). See Appendix C.1 for results on node classification graph downstream tasks.

**Q2) EXPASS relieves Oversmoothing in GNNs.** We examine the oversmoothing (using the Group Distance Ratio metric [18]) and predictive performance of GNNs trained using EXPASS with their vanilla counterparts. The oversmoothing problem in GNNs shows that the representations of nodes converge to similar vectors as the number of layers increases. Therefore, we analyze the oversmoothing of the GNNs for an increasing number of layers and find that, on average, across two architectures, EXPASS improves the group distance ratio by 34.4% (Figure 2). Further, we also analyzed the oversmoothing behavior of EXPASS for node classification tasks (in Appendix C.1) and our results indicate an inherent trade-off between oversmoothing and predictive performance of GNNs (Figures 5-7).

**Q3) Ablation studies.** We conduct ablations on several components of EXPASS with respect to its oversmoothing and predictive performance.

**EXPASS for different TopK Explanations.** We investigate the oversmoothing and predictive performance of GNNs for different topK explanations (i.e., topK edges identified by a GNN explanation) chosen in the message passing. Results show that EXPASS alleviates oversmoothing by using only the topK edges to learn graph embeddings and explicitly filter out the noise from unimportant edges. In particular, we observe that the GDR values decrease (denoting higher oversmoothing) with the increase in the use of topK edges (Figure 3). More specifically, we find that the GDR value at topK=0.1 is 11.92% higher than vanilla message passing (i.e., using all edges in the graph).

**EXPASS vs. DropEdge.** We compare the predictive and oversmoothing and predictive performance of EXPASS and DropEdge. Here, we show that message passing using optimized explanation-directed information outperforms random edge removal. We find that EXPASS outperforms DropEdge across both oversmoothing and accuracy metrics. In particular, on average, across different topK values, EXPASS improves the oversmoothing, AUROC, and F1-score performance of vanilla message passing by 71.16%, 9.53%, and 12.63%, respectively (Figure 3).

**EXPASS using Node Explanations.** We investigate the effect of the choice of the baseline explanation method on the performance of EXPASS with respect to the vanilla message passing framework. More specifically, we evaluate the predictive and explainability performance of EXPASS-augmented GNNs when trained using node explanations generated using Integrated Gradients (IG) [44]. Similar to

Table 1: Results of EXPASS for five GNNs and four graph datasets. Shown is average performance across three independent runs. Arrows ($\uparrow$, $\downarrow$) indicate the direction of better performance. EXPASS improves the predictive power (AUROC and F1-score) and degree of explainability (Graph Explanation Faithfulness) of original GNNs across multiple datasets (shaded area). Values corresponding to best performance are bolded.

| Dataset | Method | AUROC ($\uparrow$) | F1-score ($\uparrow$) | GEF ($\downarrow$) |
|---|---|---|---|---|
| ALKANE-CARBONYL | GCN | $0.97_{\pm 0.01}$ | $0.95_{\pm 0.01}$ | $0.33_{\pm 0.02}$ |
| | EXPASS-GCN | $\mathbf{0.98}_{\pm 0.00}$ | $\mathbf{0.96}_{\pm 0.01}$ | $\mathbf{0.23}_{\pm 0.02}$ |
| | GraphConv | $0.97_{\pm 0.01}$ | $0.94_{\pm 0.00}$ | $0.38_{\pm 0.05}$ |
| | EXPASS-GraphConv | $\mathbf{0.98}_{\pm 0.00}$ | $\mathbf{0.97}_{\pm 0.00}$ | $\mathbf{0.22}_{\pm 0.03}$ |
| | LeConv | $0.98_{\pm 0.01}$ | $0.96_{\pm 0.00}$ | $0.37_{\pm 0.03}$ |
| | EXPASS-LeConv | $0.98_{\pm 0.00}$ | $0.96_{\pm 0.01}$ | $\mathbf{0.24}_{\pm 0.03}$ |
| | GraphSAGE | $0.98_{\pm 0.00}$ | $0.96_{\pm 0.00}$ | $0.40_{\pm 0.12}$ |
| | EXPASS-GraphSAGE | $\mathbf{0.99}_{\pm 0.00}$ | $\mathbf{0.97}_{\pm 0.01}$ | $\mathbf{0.18}_{\pm 0.06}$ |
| | GIN | $0.96_{\pm 0.01}$ | $0.94_{\pm 0.02}$ | $0.35_{\pm 0.06}$ |
| | EXPASS-GIN | $\mathbf{0.98}_{\pm 0.01}$ | $\mathbf{0.96}_{\pm 0.02}$ | $\mathbf{0.11}_{\pm 0.04}$ |
| DD | GCN | $0.73_{\pm 0.02}$ | $0.70_{\pm 0.02}$ | $0.49_{\pm 0.04}$ |
| | EXPASS-GCN | $\mathbf{0.74}_{\pm 0.01}$ | $0.70_{\pm 0.02}$ | $\mathbf{0.30}_{\pm 0.09}$ |
| | GraphConv | $0.75_{\pm 0.03}$ | $0.73_{\pm 0.03}$ | $0.25_{\pm 0.10}$ |
| | EXPASS-GraphConv | $\mathbf{0.77}_{\pm 0.03}$ | $0.73_{\pm 0.03}$ | $\mathbf{0.19}_{\pm 0.04}$ |
| | LeConv | $0.76_{\pm 0.03}$ | $\mathbf{0.74}_{\pm 0.02}$ | $\mathbf{0.17}_{\pm 0.03}$ |
| | EXPASS-LeConv | $\mathbf{0.77}_{\pm 0.03}$ | $0.73_{\pm 0.04}$ | $0.31_{\pm 0.10}$ |
| | GraphSAGE | $0.74_{\pm 0.02}$ | $0.70_{\pm 0.02}$ | $0.21_{\pm 0.04}$ |
| | EXPASS-GraphSAGE | $\mathbf{0.76}_{\pm 0.03}$ | $\mathbf{0.71}_{\pm 0.02}$ | $\mathbf{0.20}_{\pm 0.03}$ |
| | GIN | $0.74_{\pm 0.01}$ | $0.70_{\pm 0.01}$ | $0.37_{\pm 0.03}$ |
| | EXPASS-GIN | $\mathbf{0.76}_{\pm 0.01}$ | $\mathbf{0.74}_{\pm 0.01}$ | $\mathbf{0.35}_{\pm 0.05}$ |
| MUTAG | GCN | $0.71_{\pm 0.11}$ | $0.87_{\pm 0.01}$ | $0.09_{\pm 0.03}$ |
| | EXPASS-GCN | $\mathbf{0.77}_{\pm 0.02}$ | $\mathbf{0.89}_{\pm 0.00}$ | $\mathbf{0.04}_{\pm 0.01}$ |
| | GraphConv | $0.91_{\pm 0.02}$ | $0.94_{\pm 0.02}$ | $0.66_{\pm 0.03}$ |
| | EXPASS-GraphConv | $\mathbf{0.93}_{\pm 0.01}$ | $\mathbf{0.94}_{\pm 0.01}$ | $\mathbf{0.24}_{\pm 0.03}$ |
| | LeConv | $0.92_{\pm 0.03}$ | $0.94_{\pm 0.02}$ | $0.65_{\pm 0.05}$ |
| | EXPASS-LeConv | $0.92_{\pm 0.03}$ | $\mathbf{0.96}_{\pm 0.01}$ | $\mathbf{0.30}_{\pm 0.06}$ |
| | GraphSAGE | $0.76_{\pm 0.02}$ | $0.86_{\pm 0.03}$ | $0.24_{\pm 0.08}$ |
| | EXPASS-GraphSAGE | $0.76_{\pm 0.02}$ | $\mathbf{0.87}_{\pm 0.03}$ | $\mathbf{0.11}_{\pm 0.03}$ |
| | GIN | $0.92_{\pm 0.02}$ | $0.93_{\pm 0.01}$ | $0.61_{\pm 0.05}$ |
| | EXPASS-GIN | $\mathbf{0.94}_{\pm 0.02}$ | $\mathbf{0.95}_{\pm 0.01}$ | $\mathbf{0.32}_{\pm 0.04}$ |
| PROTEINS | GCN | $0.73_{\pm 0.05}$ | $0.68_{\pm 0.04}$ | $0.19_{\pm 0.02}$ |
| | EXPASS-GCN | $\mathbf{0.74}_{\pm 0.03}$ | $\mathbf{0.69}_{\pm 0.03}$ | $\mathbf{0.08}_{\pm 0.02}$ |
| | GraphConv | $0.75_{\pm 0.03}$ | $0.70_{\pm 0.03}$ | $0.49_{\pm 0.06}$ |
| | EXPASS-GraphConv | $0.75_{\pm 0.03}$ | $0.70_{\pm 0.04}$ | $\mathbf{0.10}_{\pm 0.03}$ |
| | LeConv | $\mathbf{0.77}_{\pm 0.03}$ | $\mathbf{0.72}_{\pm 0.04}$ | $0.51_{\pm 0.01}$ |
| | EXPASS-LeConv | $0.76_{\pm 0.02}$ | $0.71_{\pm 0.03}$ | $\mathbf{0.15}_{\pm 0.07}$ |
| | GraphSAGE | $0.73_{\pm 0.04}$ | $0.69_{\pm 0.04}$ | $0.17_{\pm 0.07}$ |
| | EXPASS-GraphSAGE | $0.73_{\pm 0.04}$ | $0.69_{\pm 0.04}$ | $\mathbf{0.06}_{\pm 0.01}$ |
| | GIN | $0.77_{\pm 0.04}$ | $0.73_{\pm 0.05}$ | $0.20_{\pm 0.07}$ |
| | EXPASS-GIN | $\mathbf{0.78}_{\pm 0.03}$ | $0.73_{\pm 0.04}$ | $\mathbf{0.19}_{\pm 0.01}$ |

the results of EXPASS with GNNExplainer as the baseline explanation method (Table 1), we find that EXPASS trained using IG explanations also improves the AUROC (+2.80%), F1-score (+1.11%), and GEF (+23.67%) of the vanilla GNN model. Our results show that the choice of explainer can make a difference in the EXPASS performance, depending on the dataset. For instance, IG is a node-masking explainer that is not considered a strong explanation method and its effects are variable across datasets [33]. We recommend using graph-specific explainers that optimize for fidelity and sparsity on the edges of the input graph, which would be a best fit to increase the performance of the network. See Appendix C.3 for results using PGMExplainer. Further, our results show that EXPASS is a model- and explainer-agnostic framework that can improve the downstream task and explainability performance across different GNN architectures using diverse GNN explainers.

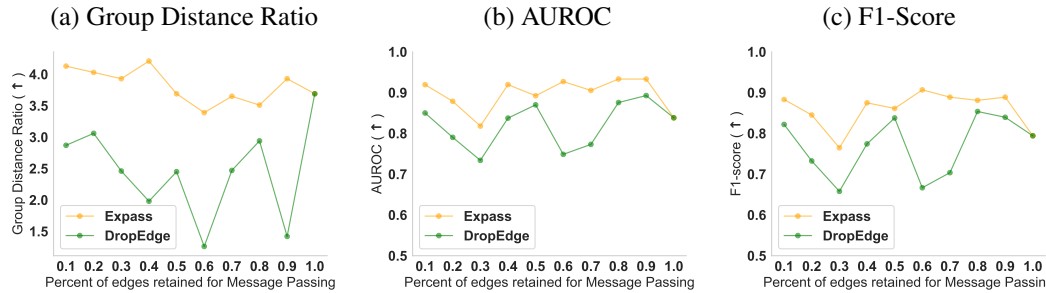

Figure 3: The effects of choosing only the topK percent of important edges on the (a) oversmoothing, (b) AUROC, and (c) F1-score performance of GCN model trained on Alkane-Carbonyl dataset. Over a wide range of topK values ($0.1 <$ topK $< 1.0$), EXPASS outperforms DropEdge [45] on all the three metrics. Note that their performance converges for topK $= 1.0$ as that denotes using all the edges in the graph.

Table 2: Results of EXPASS for GCN using the node explanations from Integrated Gradients [44] for message passing for various datasets. Shown is average performance across three independent runs. Arrows ($\uparrow, \downarrow$) indicate the direction of better performance. EXPASS improves the predictive power (AUROC and F1-score) and degree of explainability (Graph Explanation Faithfulness) of original GNNs across multiple datasets (shaded area).

| Dataset | Method | AUROC ($\uparrow$) | F1-score ($\uparrow$) | GEF ($\downarrow$) |
|---|---|---|---|---|
| DD | GCN | $0.73_{\pm 0.02}$ | $0.70_{\pm 0.02}$ | $0.25_{\pm 0.03}$ |
| | EXPASS-GCN | $\mathbf{0.75}_{\pm 0.01}$ | $\mathbf{0.71}_{\pm 0.03}$ | $\mathbf{0.23}_{\pm 0.04}$ |
| ALKANE | GCN | $0.97_{\pm 0.01}$ | $0.95_{\pm 0.01}$ | $\mathbf{0.09}_{\pm 0.01}$ |
| | EXPASS-GCN | $0.97_{\pm 0.01}$ | $0.95_{\pm 0.01}$ | $0.1_{\pm 0.01}$ |
| MUTAG | GCN | $0.71_{\pm 0.11}$ | $0.87_{\pm 0.01}$ | $0.09_{\pm 0.02}$ |
| | EXPASS-GCN | $\mathbf{0.77}_{\pm 0.02}$ | $\mathbf{0.88}_{\pm 0.01}$ | $\mathbf{0.04}_{\pm 0.02}$ |
| PROTEINS | GCN | $0.73_{\pm 0.04}$ | $\mathbf{0.68}_{\pm 0.04}$ | $0.05_{\pm 0.01}$ |
| | EXPASS-GCN | $0.73_{\pm 0.04}$ | $0.67_{\pm 0.05}$ | $\mathbf{0.04}_{\pm 0.01}$ |

**EXPASS vs. Attention.** We demonstrate the utility of using explanations vs. attention weights in the message passing step using GAT [43] model architecture. On average, across four datasets, we find that EXPASS achieves higher AUROC (+3.85%) and F1-score (+2.24%) than the attention-based GAT model (Table 3). In addition, GNNExplainer [13] demonstrated that post hoc GNN explainers generate better explanations than attention weights, which further highlights the benefits of EXPASS. In comparison to EXPASS, GAT can be considered as a special case of our framework, where attention weights replace explanations. On the other hand, EXPASS has larger benefits since it can be applied to any existing GNN architectures that lack explainability.

**Q4) Visualizing explanations.** Here, we visualize how the explanation develops over the training process of the GNN model. In particular, we visualize the generated explanations from EXPASS-GCN with GNNExplainer trained on the MUTAG dataset at different epochs during the training process and find that the explanations converge to the ground-truth explanation of a non-mutagenic molecule (i.e., the absence of a carbon ring alongside the highlighted $NO_2$ molecules) as the training progresses (Figure 8). Further, we compare the generated explanations for a vanilla GCN and its EXPASS counterpart and find that the explanation for vanilla GCN falsely identifies the carbon-carbon bonds as important (Figure 4). This qualitative analysis provides further evidence for the observed higher faithfulness results (Table 1) of explanations generated using our proposed EXPASS framework.

## 6 Conclusion and Discussion

In this work, we propose the problem of learning graph embeddings using explanation-directed message passing in GNNs. To this end, we introduce EXPASS, a novel message-passing framework that can be used with any existing GNN model and subgraph-optimizing explainer to learn accurate embeddings by aggregating only embeddings from nodes and edges identified as important by a GNN explainer. We perform an extensive theoretical analysis to show that EXPASS relieves the oversmoothing problem in GNNs, and the embedding difference between the vanilla message passing

Table 3: Results of EXPASS and GAT for various datasets. Shown is the average performance across three independent runs. Arrows (↑, ↓) indicate the direction of better performance. EXPASS improves the predictive power (AUROC and F1-score) and degree of explainability (Graph Explanation Faithfulness) of original GNNs across multiple datasets (shaded area).

| Dataset | Method | AUROC (↑) | F1-score (↑) |
|---|---|---|---|
| DD | GAT | $0.72_{\pm0.02}$ | $0.68_{\pm0.03}$ |
|  | EXPASS-GCN | $\mathbf{0.74}_{\pm0.01}$ | $\mathbf{0.70}_{\pm0.01}$ |
| ALKANE | GAT | $0.97_{\pm0.01}$ | $0.95_{\pm0.01}$ |
|  | EXPASS-GCN | $\mathbf{0.98}_{\pm0.00}$ | $\mathbf{0.96}_{\pm0.01}$ |
| MUTAG | GAT | $0.69_{\pm0.10}$ | $0.86_{\pm0.01}$ |
|  | EXPASS-GCN | $\mathbf{0.77}_{\pm0.02}$ | $\mathbf{0.89}_{\pm0.00}$ |
| PROTEINS | GAT | $0.74_{\pm0.04}$ | $0.68_{\pm0.04}$ |
|  | EXPASS-GCN | $0.74_{\pm0.03}$ | $\mathbf{0.69}_{\pm0.03}$ |

VanillaGCN                                             EXPASS-GCN

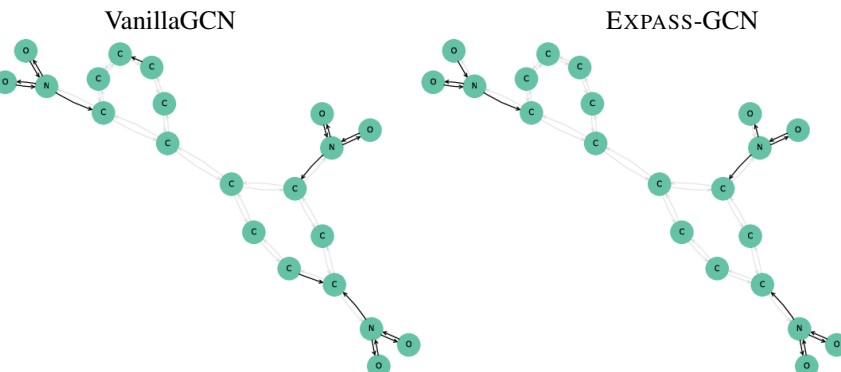

Figure 4: Visualizing the explanation generated for a non-mutagenic molecule prediction using Vanilla GCN (left) and EXPASS-GCN (right) with GNNExplainer method. Note that the explanation from vanilla GCN falsely identifies the carbon-carbon bonds as important. This qualitative analysis provides further evidence for the observed higher faithfulness results of explanations generated using our proposed EXPASS framework.

framework and EXPASS can be upper bounded by the difference of their respective layer weights. Our empirical results on benchmark datasets show that EXPASS improves the explainability of the underlying GNN model without sacrificing its predictive performance. However, the training of EXPASS depends on the choice of explanation method, the number of data points to explain, and the dataset of choice, which is computationally more expensive than its vanilla counterparts. We find that the training time of EXPASS can be improved by using techniques like batch processing and efficient sampling of correctly-classified nodes and graphs. Further, adapting post-hoc explainers to generate subgraphs utilizing the embedding space would also improve the computation time of EXPASS. Our proposed method and findings open exciting new avenues to learn graph representations by jointly training models and explanation methods. We anticipate that EXPASS could open new frontiers in graph machine learning for developing explanation-based training frameworks.

## Acknowledgements

The authors would like to thank the anonymous reviewers for their helpful feedback that helped improve the work. CA would like to thank Lasse Mohr and Samuele Firmani for the helpful discussions at the beginning of the project and LOGML Summer School for connecting with the students. The views expressed here are those of the authors and do not reflect the official policy or position of the affiliated company.

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

# A    Proofs for Theorems in Section 4

**Theorem 1.** *Given a non-linear activation function $\sigma$ that is Lipschitz continuous, the difference between the node embeddings between a vanilla message passing and* EXPASS *framework can be bounded by the difference in their individual weights, i.e.,*

$$\|\mathbf{h}_u^{(l)} - \mathbf{h'}_u^{(l)}\|_2 \leq \|\mathbf{W}_a^{(l)} - \mathbf{W'}_a^{(l)}\|_2 \|\mathbf{h}_u^{(l-1)}\|_2 + \|\mathbf{W}_n^{(l)} - \mathbf{W'}_n^{(l)}\|_2 \sum_{v \in \mathcal{N}_u \cap s_v = 1} \|\mathbf{h}_v^{(l-1)}\|_2, \tag{3}$$

*where $\mathbf{W}_a^{(l)}$ and $\mathbf{W'}_a^{(l)}$ are the weights for node $u$ in layer $l$ of the vanilla message passing and* EXPASS *framework and $\mathbf{W}_n^{(l)}$ and $\mathbf{W'}_n^{(l)}$ are their respective weight matrix with the neighbors of node $u$ at layer $l$.*

*Proof.*  For a given node $u$, the node representation output by layer $l$ of the GNN is given by:

$$\mathbf{h}_u^{(l)} = \sigma\Big(\mathbf{W}_a^{(l)} \mathbf{h}_u^{(l-1)} + \mathbf{W}_n^{(l)} \sum_{v \in \mathcal{N}_u} \mathbf{h}_v^{(l-1)}\Big), \tag{4}$$

where we consider the AGG operator as a fully-connected layer, UPD to be a sigmoid activation function $\sigma(\cdot)$, $\mathbf{W}_a^{(l)}$ is the weights for node $u$ in layer $l$ and $\mathbf{W}_n^{(l)}$ is the weight matrix with the neighbors of node $u$ at layer $l$.

Let us consider an edge *in-hoc* explanation that generates a binary mask highlighting the important edges for the prediction of node $u$. Note that using the edge mask, we can also get a node-level mask signifying the importance of neighboring nodes. Let us denote that node explanation mask as $s_v$ where $s_v = 1$ if the node is important, otherwise $s_v = 0$. Formally, the corresponding message passing equations for EXPASS can be written as:

$$\mathbf{h'}_u^{(l)} = \sigma\Big(\mathbf{W'}_a^{(l)} \mathbf{h'}_u^{(l-1)} + \mathbf{W'}_n^{(l)} \sum_{v \in \mathcal{N}_u} s_v \mathbf{h'}_v^{(l-1)}\Big), \tag{5}$$

where $\mathbf{h'}_u^{(l)}$ and $\mathbf{h'}_v^{(l)}$ represents the embeddings of node $u$ and $v$ using the feedback explanation, and $\mathbf{W'}_a^{(l)}$ and $\mathbf{W'}_n^{(l)}$ represents the corresponding weights at layer $l$ for GNN model trained using EXPASS.

The difference between the node embeddings obtained after the message-passing in layer $l$ from Equations 4-5 is given as:

$$\mathbf{h}_u^{(l)} - \mathbf{h'}_u^{(l)} = \sigma\Big(\mathbf{W}_a^{(l)} \mathbf{h}_u^{(l-1)} + \mathbf{W}_n^{(l)} \sum_{v \in \mathcal{N}_u} \mathbf{h}_v^{(l-1)}\Big) - \sigma\Big(\mathbf{W'}_a^{(l)} \mathbf{h'}_u^{(l-1)} + \mathbf{W'}_n^{(l)} \sum_{v \in \mathcal{N}_u} s_v \mathbf{h'}_v^{(l-1)}\Big), \tag{6}$$

Taking the $\ell_2$-norm on both sides and assuming a normalized Lipschitz non-linear sigmoid activation, *i.e.,* $\|\sigma(b) - \sigma(a)\|_2 \leq \|b - a\|_2$, we get:

$$\|\mathbf{h}_u^{(l)} - \mathbf{h'}_u^{(l)}\|_2 = \|\sigma\Big(\mathbf{W}_a^{(l)} \mathbf{h}_u^{(l-1)} + \mathbf{W}_n^{(l)} \sum_{v \in \mathcal{N}_u} \mathbf{h}_v^{(l-1)}\Big) - \sigma\Big(\mathbf{W'}_a^{(l)} \mathbf{h'}_u^{(l-1)} + \mathbf{W'}_n^{(l)} \sum_{v \in \mathcal{N}_u} s_v \mathbf{h'}_v^{(l-1)}\Big)\|_2$$

$$\leq \|\mathbf{W}_a^{(l)} \mathbf{h}_u^{(l-1)} + \mathbf{W}_n^{(l)} \sum_{v \in \mathcal{N}_u} \mathbf{h}_v^{(l-1)} - \mathbf{W'}_a^{(l)} \mathbf{h'}_u^{(l-1)} - \mathbf{W'}_n^{(l)} \sum_{v \in \mathcal{N}_u} s_v \mathbf{h'}_v^{(l-1)}\|_2$$

$$\leq \|\mathbf{W}_a^{(l)} \mathbf{h}_u^{(l-1)} - \mathbf{W'}_a^{(l)} \mathbf{h'}_u^{(l-1)} + \mathbf{W}_n^{(l)} \sum_{v \in \mathcal{N}_u} \mathbf{h}_v^{(l-1)} - \mathbf{W'}_n^{(l)} \sum_{v \in \mathcal{N}_u} s_v \mathbf{h'}_v^{(l-1)}\|_2$$

$$\leq \|\mathbf{W}_a^{(l)} \mathbf{h}_u^{(l-1)} - \mathbf{W'}_a^{(l)} \mathbf{h}_u^{(l-1)}\|_2 + \|\mathbf{W}_n^{(l)} \sum_{v \in \mathcal{N}_u \cap s_v = 0} \mathbf{h}_v^{(l-1)} + (\mathbf{W}_n^{(l)} - \mathbf{W'}_n^{(l)}) \sum_{v \in \mathcal{N}_u \cap s_v = 1} \mathbf{h}_v^{(l-1)}\|_2$$

(Using Triangle Inequality and Faithfulness property of explanations)

Given a faithful explanation, the node embeddings for node $u$ using the vanilla message passing network are equivalent to that EXPASS since most explainers optimize the mask to approximate the input embedding. More specifically, for a given node embedding $\mathbf{h'}_u^{(l-1)} = \mathbf{h}_u^{(l-1)} + \epsilon_u$, a faithful explanation bounds the $\epsilon_u$ to zero. In addition to faithfulness, a GNN using vanilla message passing

and EXPASS can predict a node $u$ to the same class only if both frameworks generate similar node embeddings (Proposition 1 in Agarwal et al. [4]).

Using Matrix-norm and Triangle Inequality for the sum in the neighborhood, we get:

$$\|\mathbf{h}_u^{(l)} - \mathbf{h}'_u^{(l)}\|_2 \leq \|(\mathbf{W}_a^{(l)} - \mathbf{W}'_a^{(l)})\,\mathbf{h}_u^{(l-1)}\|_2 + \|\mathbf{W}_n^{(l)}\|_2 \sum_{v \in \mathcal{N}_u \cap s_v = 0} \|\mathbf{h}_v^{(l-1)}\|_2 +$$
$$\|\mathbf{W}_n^{(l)} - \mathbf{W}'_n^{(l)}\|_2 \sum_{v \in \mathcal{N}_u \cap s_v = 1} \|\mathbf{h}_v^{(l-1)}\|_2$$

Again, using the faithfulness property of explanations, the contribution of node embeddings from node $v \in \mathcal{N}(u) \cap s_v = 0$ is irrelevant to the final embedding and can be removed. Finally, using Matrix-norm inequality on the first term, we get:

$$\|\mathbf{h}_u^{(l)} - \mathbf{h}'_u^{(l)}\|_2 \leq \|\mathbf{W}_a^{(l)} - \mathbf{W}'_a^{(l)}\|_2 \|\mathbf{h}_u^{(l-1)}\|_2 + \|\mathbf{W}_n^{(l)} - \mathbf{W}'_n^{(l)}\|_2 \sum_{v \in \mathcal{N}(u) \cap s_v = 1} \|\mathbf{h}_v^{(l-1)}\|_2$$

Thus, we observe that the embedding difference at layer $l$ between a vanilla message passing network and the EXPASS is purely based on the difference between their weights and the embeddings of node $u$ and its subgraph. □

**Definition 2** (Dirichlet Energy for a Node Embedding Matrix [35]). *Given a node embedding matrix* $\mathbf{h}^{(l)} = [\mathbf{h}_1^{(l)}, \ldots, \mathbf{h}_n^{(l)}]^T$ *learned from the GNN model at the $l^{th}$ layer, the Dirichlet Energy* $E(\mathbf{h}^{(l)})$ *is defined as:*

$$E(\mathbf{h}^{(l)}) = tr(\mathbf{h}^{(l)^T} \tilde{\Delta} \mathbf{h}^{(l)}) = \frac{1}{2} \sum_{i,j \in \mathcal{V}} a_{ij} \|\frac{\mathbf{h}_i^{(l)}}{\sqrt{1+d_i}} - \frac{\mathbf{h}_j^{(l)}}{\sqrt{1+d_j}}\|_2^2 \tag{7}$$

*where* $a_{ij}$ *are elements in the adjacency matrix* $\tilde{\mathbf{A}}$ *and* $d_i, d_j$ *is the degree of node* $i$ *and* $j$*, respectively.*

Cai et al. [34] extensively show that higher Dirichlet energies correspond to lower oversmoothing. Furthermore, they show that the removal of edges or ,similarly, reduction of edge weights on graphs help alleviate oversmoothing.

**Proposition 1 (EXPASS relieves Oversmoothing).** EXPASS *alleviates oversmoothing by slowing the layer-wise loss of Dirichlet energy.*

*Proof Sketch.* Here, we show the capabilities of EXPASS as a framework that alleviates the oversmoothing problem in GNNs. To this end, we utilize the bounds on the Dirichlet energy of the EXPASS embeddings at the $l^{th}$ layer of the GNN model by Zhou et al. [35]:

$$(1 - \lambda_1)^2 s_{min}^{(l)} E(\mathbf{h}^{(l-1)}) \leq E(\mathbf{h}^{(l)}) \leq (1 - \lambda_0)^2 s_{max}^{(l)} E(\mathbf{h}^{(l-1)}), \tag{8}$$

where $\lambda_1, \lambda_0$ are the non-zero eigenvalues of the symmetric normalized Laplacian $\tilde{\Delta}$ that is closest to 1 and 0, respectively, and $s_{min}^{(l)}, s_{max}^{(l)}$ are the squares of the minimum and maximum singular values of weight $\mathbf{W}^{(l)}$, respectively. Since EXPASS reduces the input graph to its specific explanation, we argue that it can alleviate oversmoothing by reducing the information propagation along irrelevant nodes and edges. From the perspective of Dirichlet energy, we know from [19] that, for Erdős-Rényi graphs, $\lambda_0$ converges to 1 as the graph becomes denser. Oono et al. [19] state that GNNs oversmooth on sufficiently large graphs (similar to Erdős-Rényi graphs). Under this assumption, EXPASS, by definition introduces sparsity inside the $\tilde{\Delta}$ of the input graph by using a smaller set of topK important edges for learning embeddings and, thus, reduces $\lambda_0$ to tighten the upper-bound in Equation 8. In practice, the choice of explainer used in EXPASS can reduce $\lambda_0$ to varying degrees. More specifically, explainers that promote sparsity would push $\lambda_0$ closer to zero and slow down the decrease of Dirichlet energy in subsequent GNN layers. Finally, we know from Cai et al. [34] that higher values of Dirichlet energy per layer correspond to lower oversmoothing, we assert that EXPASS alleviates oversmoothing. □

# B Experiment

## B.1 Datasets

**Mutag.** The MUTAG [36] dataset contains 188 graph molecules labeled into two different classes according to their mutagenic properties, i.e., effect on the *Gram-negative bacterium S. Typhimuriuma*. Kazius et al. [36] identifies several toxicophores - motifs in the molecular graph - that correlate with mutagenicity.

**Alkane-Carbonyl.** The Alkane-Carbonyl [37] dataset contains 1,125 molecular graphs categorized into two classes where an instance in the positive group indicates a molecule that contains an unbranched alkane and a carbonyl (C=O) functional group.

**DD.** The DD [38] dataset was derived from [46] and contains 1,178 protein graphs where nodes represent individual amino-acids and edges represent their spatial proximity. The task is to predict whether a given protein is an enzyme or not.

**Proteins.** The Proteins [39] dataset was derived from [46] and contains 1,113 protein graphs where nodes represent secondary structure elements and edges indicate neighborhood in the amino-acid sequence or the 3D space. The task is to predict whether a given protein is an enzyme or not.

**PubMed.** The PubMed dataset [47] is a citation network from the PubMed database, with over 4 million nodes and edges respectively. It contains a bag-of-words representation of documents and citation links between documents. The task is to predict a node's class among 3 classes.

## B.2 Implementation details

**GNN libraries and models.** All our models were implemented using PyTorch Geometric (2.1.0) and PyTorch (1.11.0). For our experiments, we used baseline GNN architectures with three layers followed by ReLU layers and set the hidden dimensionality to 32. Finally, we used a single linear layer to transform the graph embeddings to their respective classes. We selected Adam as our optimizer and a weighted Cross Entropy Loss to train both vanilla and EXPASS frameworks. All models were trained over three independent runs with a learning rate of 0.01 for 200 epochs for DD and Proteins datasets and 150 epochs for Alkane and MUTAG datasets.

**EXPASS.** We define the burn-in period as a number of epochs during training in which no explanations are used. The burn-in period is necessary to avoid feeding spurious explanations to the model since an untrained model can lead to unfaithful explanations. The length of the burn-in period was treated as a hyperparameter and fine-tuned during the model fine-tuning phase. After fine-tuning, we found that a burn-in period in the range [5, 15] worked best, whereas most EXPASS models outperformed their vanilla counterparts using a burn-in period of 5 and 10 epochs in our experiments. We generated explanations for a specific percentage of correctly predicted graphs sampled in each batch and were set to 0.4 for all our experiments. The generated explanations are normalized to [0, 1] and hard-masked over the topK most relevant edges, where topK is a percentage of the total number of edges in the input graph and was set to topK $\in [0.3, 0.4]$ for our experiments.

**GNN explanation methods.** At each epoch, the model weights were frozen to generate explanations, which were calculated as the median over *n* independent runs of GNNExplainer, in order to obtain consistent explanations. Then, the model weights were trained using generated explanations. We chose the median instead of the mean to prevent the individual outliers from significantly changing the final explanations. The number of individual runs of an explainer was treated as a hyperparameter and was set to five for GNNExplainer and one for Integrated Gradients (as it generates consistent explanations over multiple runs). In each run, the GNNExplainer was trained for 200 epochs (150 in the case of Alkane) with a learning rate of 0.01. All other hyperparameters of the explanation methods were set using the author's guidelines. Note that these multiple iterations of the explainers are not required for EXPASS to perform well when using other stable GNN explainers. To summarize, the following parameters were treated as hyperparameters: the learning rate of the model, the learning rate of the explanation method, the number of epochs the explanation method was trained for, the number of times the GNNExplainer was computed at each epoch for each sampled graph, the percentage of correctly classified graphs that were randomly sampled to compute the explanations and the percentage of top edges/nodes that were selected as the most relevant. On the other hand, in the case of vanilla models, the learning rate was fine-tuned during the tuning phase.

**Dataset.** The train, validation, and test split was at 80%, 10%, and 10% for Alkane, Proteins, and DD following prior works. In the case of MUTAG, no validation set was used due to the smaller dataset size, and the train and test split was at 80% and 20%.

**GNN performance metric.** The GEF scores were evaluated as the mean over the individual scores of all generated explanations on the test dataset, where the explanations were hard-masked with topK $= 0.1$ for GNNExplainer, and topK $= 0.25$ for Integrated Gradients/PGMExplainer. Note that, since Integrated Gradients/PGMExplainer generates a node mask instead of an edge mask, we required a higher topK value to generate a non-empty hard mask over the input graphs, since we retain the topK most relevant nodes in the explanation mask.

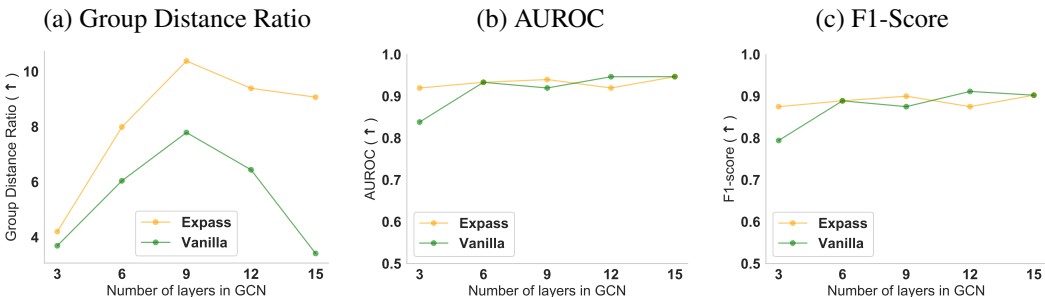

Figure 5: The effects of the number of GNN layers on the (a) oversmoothing, (b) AUROC, and (c) F1-score performance of EXPASS-GCN and Vanilla-GCN trained on Alkane-Carbonyl dataset. Across models with increasing number of layers, EXPASS achieves higher GDR performance without sacrificing the predictive performance of the GCN model.

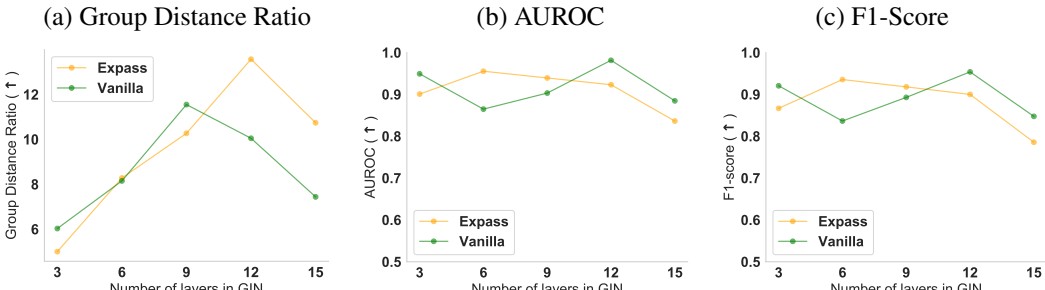

Figure 6: The effects of the number of GNN layers on the (a) oversmoothing, (b) AUROC, and (c) F1-score performance of EXPASS-GIN and Vanilla-GIN trained on Alkane-Carbonyl dataset. We observe that, across models with an increasing number of layers, EXPASS achieves higher GDR performance and there exists an inherent trade-off between oversmoothing and predictive performance of GIN.

## C Additional results

### C.1 Node classification results

We extend our proposed framework to GNN models trained on different graph downstream tasks. In particular, we conduct additional experiments to obtain the over-smoothing and predictive behavior of EXPASS for node-level tasks. We train five state-of-the-art GNN models and their EXPASS counterparts on the PubMed node classification dataset. Our results show that EXPASS alleviates the over-smoothing effect in GNNs for models with higher depths (Figure 7) and achieves on-par or higher predictive performance (Table 4). We find that, on average, EXPASS augmented GCN achieves 19.53% better over-smoothing performance for node-classification GNN models with higher depths.

### C.2 Burn-in period

The burn-in period was treated as a hyperparameter and fine-tuned for each dataset and architecture. An example of the effect of the burn-in period on the AUROC and F1-score is reported in Table 5, where the change in performance is evaluated for the Proteins dataset when using a lag of 5, 10, and 15.

Table 4: Results of EXPASS for five GNNs using PubMed node classification dataset. Shown is the average performance across five independent runs. Arrows (↑, ↓) indicate the direction of better performance. EXPASS improves the predictive power (testing accuracy) of original GNNs across multiple datasets (shaded area).

| Method | Testing Accuracy (↑) |
|---|---|
| GCN | $0.7596_{\pm 0.002}$ |
| EXPASS-GCN | $\mathbf{0.7616}_{\pm 0.002}$ |
| GraphConv | $0.7652_{\pm 0.002}$ |
| EXPASS-GraphConv | $\mathbf{0.7682}_{\pm 0.002}$ |
| LeConv | $0.7424_{\pm 0.003}$ |
| EXPASS-LeConv | $0.7244_{\pm 0.010}$ |
| GraphSAGE | $0.7462_{\pm 0.004}$ |
| EXPASS-GraphSAGE | $\mathbf{0.7533}_{\pm 0.002}$ |
| GIN | $0.7233_{\pm 0.002}$ |
| EXPASS-GIN | $\mathbf{0.7310}_{\pm 0.009}$ |

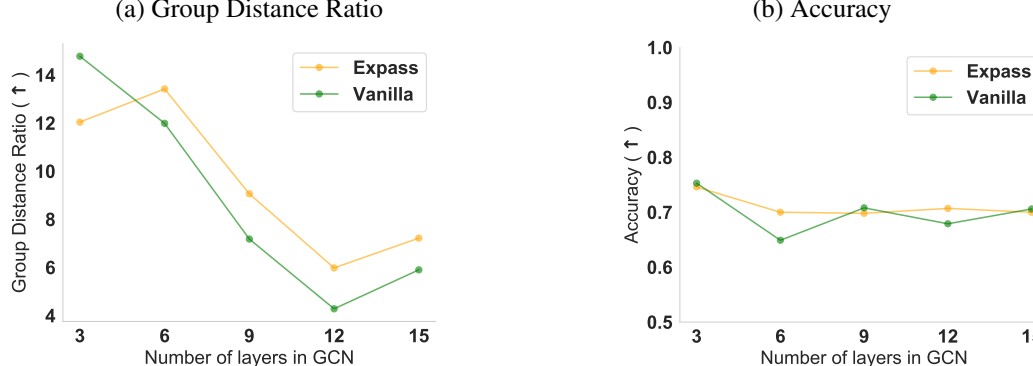

(a) Group Distance Ratio         (b) Accuracy

Figure 7: The effects of the number of GNN layers on the (a) oversmoothing, (b) testing accuracy performance of EXPASS-GCN and Vanilla-GCN trained on PubMed node classification dataset. We observe that, across models with an increasing number of layers, EXPASS achieves higher GDR performance and achieves on-par or better testing accuracy.

## C.3 PGMExplainer results

Here, we showcase the flexibility of the EXPASS with respect to different GNN explainers. In Table 6, we show the effectiveness of EXPASS with respect to different explainers by further utilizing PGMExplainer [12] as the explanation generator. We utilize the graph explanations of PGMExplainer as node masks on the input graph (as they cannot generate edge-level masks) and incorporate them using our explanation-aware message-passing scheme with GCN as our underlying architecture. On average, across three datasets, we find that EXPASS trained using PGM-Explainer achieves higher GEF (+36.56%) than their vanilla counterparts (Table 6). Also, it achieves a boost of 11.27% in AUROC for MUTAG and a 1% improvement in the F1-score for the Alkane dataset. We observe that PGMExplainer, similar to Integrated Gradients, produces node masks, which lack detail and do not provide finer changes to the underlying GNN model, like edge masks. We hypothesize that this contributes to the large variation in the predictive performance across datasets.

Table 5: Results of EXPASS for various burn-in periods. Shown is the average performance across three independent runs (and standard error). Arrows (↑, ↓) indicate the direction of better performance.

| Method | Burn-in period | AUROC (↑) | F1-score (↑) |
|---|---|---|---|
| EXPASS-GIN | 5 | $0.76 \pm 0.04$ | $0.72 \pm 0.05$ |
| | 10 | $\mathbf{0.78} \pm 0.03$ | $\mathbf{0.73} \pm 0.04$ |
| | 15 | $\mathbf{0.78} \pm 0.03$ | $\mathbf{0.73} \pm 0.03$ |
| EXPASS-GraphSAGE | 5 | $0.73 \pm 0.04$ | $0.68 \pm 0.04$ |
| | 10 | $0.73 \pm 0.04$ | $\mathbf{0.69} \pm 0.04$ |
| | 15 | $0.73 \pm 0.04$ | $\mathbf{0.69} \pm 0.04$ |
| EXPASS-LeConv | 5 | $\mathbf{0.76} \pm 0.02$ | $\mathbf{0.71} \pm 0.03$ |
| | 10 | $0.74 \pm 0.04$ | $0.69 \pm 0.04$ |
| | 15 | $0.75 \pm 0.03$ | $0.71 \pm 0.03$ |

Table 6: Results of EXPASS for GCN using the node explanations from PGMExplainer [12] for message passing for various datasets. Shown is the average performance across three independent runs. Arrows (↑, ↓) indicate the direction of better performance. EXPASS improves the predictive power (AUROC and F1-score) and degree of explainability (Graph Explanation Faithfulness) of original GNNs across multiple datasets (shaded area).

| Dataset | Method | AUROC (↑) | F1-score (↑) | GEF (↓) |
|---|---|---|---|---|
| ALKANE | GCN | $0.97 \pm 0.01$ | $0.95 \pm 0.01$ | $0.31 \pm 0.02$ |
| | EXPASS-GCN | $\mathbf{0.97} \pm 0.01$ | $\mathbf{0.96} \pm 0.01$ | $\mathbf{0.28} \pm 0.03$ |
| MUTAG | GCN | $0.71 \pm 0.11$ | $\mathbf{0.87} \pm 0.01$ | $0.21 \pm 0.07$ |
| | EXPASS-GCN | $\mathbf{0.79} \pm 0.03$ | $0.86 \pm 0.01$ | $\mathbf{0.07} \pm 0.01$ |
| PROTEINS | GCN | $0.73 \pm 0.04$ | $\mathbf{0.68} \pm 0.04$ | $0.03 \pm 0.00$ |
| | EXPASS-GCN | $0.66 \pm 0.02$ | $0.67 \pm 0.05$ | $\mathbf{0.02} \pm 0.00$ |

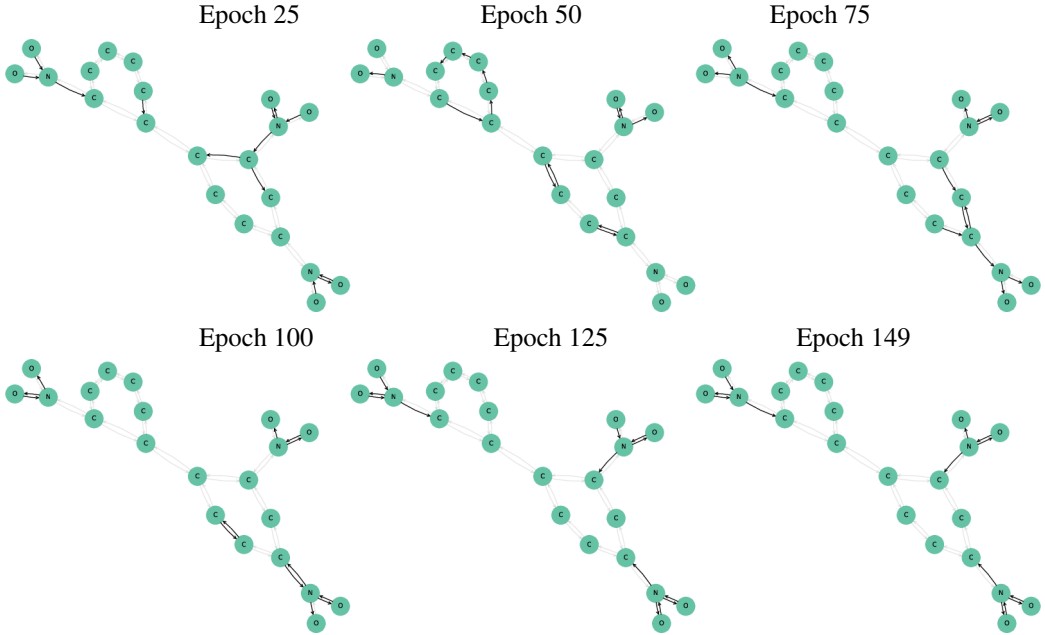

Figure 8: Generated explanations from EXPASS-GCN trained on the MUTAG dataset at different epochs during the training process and find that the explanations does converge to the ground-truth explanation of a mutagenic molecule (i.e., the absence of a carbon ring) as the training progresses. This qualitative analysis provides further evidence for the observed higher faithfulness results of explanations generated using our proposed EXPASS framework.

