# OpenReview forum: "Towards Training GNNs using Explanation Directed Message Passing"
_logconference.io/LOG/2022/Conference — LoG 2022 Poster_

### Official Review · Reviewer_JBJh · 2022-10-19

**Overall Score:** 5
**Confidence:** 4

**Review:**

This paper proposes an explanation-directed training framework for GNNs, i.e., EXPASS, for the graph representation learning. EXPASS learns accurate graph embeddings using only important nodes and edges, which are identified by GNN explanation methods. The authors theoretically and empirically show that EXPASS relieves the oversmoothing problem in GNNs by discarding unimportant nodes and edges. Experiments on several molecular chemistry datasets demonstrate that GNNs trained using EXPASS outperform vanilla GNNs in terms of AUROC and F1-score.

However, my concerns are as follows.

1. The used various techniques---including (i) generating importance scores for nodes and edges; (ii) removing the knowledge of irrelevant nodes and edges, and then aggregating messages from important ones---seem well-investigated. Specifically, GAT [1] has applied the idea of assigning an importance score for each node. Besides, the idea of masking out unimportant messages from neighbors and selecting an important subgraph for each node has been explored by GNNExplainer [2]. Although there is no related work that combines explainers with the training of GNNs, the novelty of this paper is limited. The authors may want to point out the clear differences between EXPASS and the aforementioned methods from the technical aspect.
2. The writing needs improvements.
a)	The meaning of "correctly classified nodes and graphs" in Line 155 is unclear.
b)	The explanation of the objective in Line 164 is missing.
c)	The authors provide a theoretical upper bound for the embedding differences between a vanilla message passing and the EXPASS framework in Theorem 1. However, it is necessary to further analyze what benefits this theoretical upper bound brings to EXPASS.

In addition, I have some questions about the paper.

1. The authors claim that the proposed framework improves the predictive performance of the underlying GNNs by incorporating explainers in the model training, which means that the effectiveness of EXPASS is supposed to be independent of the choice of explainers. However, this paper uses only two explanation methods---GNNExplainer and Integrated Gradients---to demonstrate the effectiveness of EXPASS. Are these sufficient to demonstrate that the performance improvements stem from the idea of EXPASS rather than the chosen explainers? The authors may want to conduct experiments with more baseline subgraph-optimizing GNN explanation methods, such as PGM-Explainer [3] and GraphMask [4].

2. The conclusion of “we find that the GDR value …” in Line 269 is confusing. How is it illustrated in Figure 3?


[1] Velickovic P, Cucurull G, Casanova A, et al. Graph attention networks.
[2] Ying Z, Bourgeois D, You J, et al. Gnnexplainer: Generating explanations for graph neural networks.
[3] Minh N Vu and My T Thai. Pgm-explainer: Probabilistic graphical model explanations for graph neural networks.
[4] Michael Sejr Schlichtkrull, Nicola De Cao, and Ivan Titov. Interpreting graph neural networks for nlp with differentiable edge masking.

---

### Official Review · Reviewer_Azpx · 2022-10-19

**Overall Score:** 6
**Confidence:** 4

**Review:**

Firstly, well done. The paper was a nice read.

Summary:
The paper proposes EXPASS, a message passing framework for GNNs that is explanation directed. The framework uses information from subgraph-optimising explainers to limit message passing to nodes and edges deemed important within the neighbourhood. The propagation of information along unimportant edges is blocked, which alleviates the issue of oversmoothing.

Strong Points:
1. The paper proposes a novel technique using explainers to produce a more explainable GNN and to mitigate oversmoothing issues.
2. The experimental set-up comprises a range of datasets and architectures. I have left suggestions for improvement below.
3. The paper is well written and good visualisations. I have left minor feedback on the writing style below.

Weak Points
1. It can be argued, that the paper lacks comparison to existing work. While important works in the field are cited, the paper could benefit from highlighting its benefits/distinctions. I am specifically referring to a comparison to Spinelli et al. (2021).
2. The paper lacks a discussion of the explainability being optimised for the explainer, which may have different drawbacks. A point of discussion could be whether the biases/beliefs of an explainer are simply reaffirmed via EXPASS.
3. It is unclear whether you retrain GNNExplainer for every training epoch, as the GNN continues to update. Could you explain this?

Questions for the authors:
1. If GNNExplainer must be retrained for every GNN update, what is the impact on the training time? Could you provide an experiment?
2. Could you clarify, if the explainer does not provide a node/edge importance score, how does the framework work? This is based on EXPASS being explainer-agnostic.

Feedback:
I think you could improve the paper by adding the following:
1. Confidence intervals for the scores reported.
2. Additional experiments on node classification tasks.
3. Mention the burn-in period earlier. It would be also interesting to have results on the optimal/recommended burn-in period in the main paper.
4. If you have time, it would be nice to see how an explanation develops over time and whether it makes more sense.
5. I think the remaining space of the paper could also be used by including the proof sketch for oversmoothing in the main section.

Feedback on Writing Style:
1. Check consistent use of post-hoc vs post hoc
2. Some sentences are very long, which makes them hard to understand. For example, the fifth sentence in the abstract.

Recommendation:
Weak Accept

Justification for Recommendation:
You have submitted a good paper with merits. While the approach is novel, I think the overall idea is similar to that of Spinelli et al. (2021) and highlighting benefits of your method would make it stand out more. I think your evaluation is good too, however, there are a few points that could be improved.

---

### Official Review · Reviewer_kvzN · 2022-10-21

**Overall Score:** 5
**Confidence:** 4

**Review:**

Summary of the paper:  Explaination GNN draws more attention in graph representation learning recently. Most works focus on Ad-hoc explaination and  lack of systematically analyzing the reliability of explaination on the model performance. This paper resorts to built-in explaination message passing mechanism into GNN training to explore the relationship between explaination and GNN performance.

Strong points: Theoretical analysis of the impact of proposed explaination messaging passing on over-smoothing problem in GNN training; and make use of metrics to quantify such impact.

Weak points: lack of deep insight of the proposed method comparing to attention guided GNNs; experiments conducted over graph classification is not an ideal case to explore the impact of over-smoothing problem.

Questions:

1.What is the difference between the proposed explained message passing with GAT (which makes use of attention score to distinguish different neighboring nodes or measure the contribution of different neighboring node to the target node)

2.Top-k selection mechanism has been proposed to improve the explainability of GNN (or GAT), please refer to Graph U-Netss. Such top-k selection mechanism has been also stated to relieve the problem of over-smoothing (Please refer to “Improving Graph Attention Networks with Large Margin-based Constraints”)

3.The problem of over-smoothing tends to learn similar node representations with deep GNNs when the nodes are connected. Therefore, such problem challenges the node classification task especially when the nodes with different class labels are connected. It is better to apply the proposed method over node classification task.

For graph-level task (such as graph classification), it is not clear that whether ‘over-smoothing’ (deep GNNs) always has a directly negative effect since: 1)  we make prediction over a single graph-level representation which is usually inferred by aggregating all node representations via the readout function; and 2) the objective is to distinguish different labels among different graphs. And over-smoothing occurs over connected nodes. Therefore, it is better to conduct experiments over node-level task; 3) Fig. 4 also shows that vanilla GCN even with 15 layers are still perform quite well but has quite low GDR.

For graph-level task, the deeper GNNs involve more parameters and might also suffer from overfitting. This might be another reason to demonstrate Fig. 5 that deeper GNNs performs worse.

Improvement:

1.It is better to conduct more experiments over node classification tasks to explore whether the proposed method can relieve over-smoothing when GNNs go deep.
2.The proposed method also emphasizes ‘explaination’ as well. It is better to give more intuitive explaination analysis (such as case study to demonstrate why explaination is helpful)

---

### Meta-Review · Area_Chair_Wm7B · 2022-11-17

**Confidence:** 4
**Recommendation:** Accept

**Meta Review:**

The paper aims to develop an explainable message passing mechanism for GNN which makes the model intrinsically explainable. Extensive discussions are exchanged between reviewers and authors. The main strength lies in the intrinsic nature of the explanation mechanism and the message passing mechanism's ability to be compatible with any GNN explanation method. The main concern by reviewer kvzN is that how to make sure the better generalization is due to the explanation, which has been properly addressed by the authors (no further feedback from the reviewer). Reviewer JBJh raised the recommendation score to weak accept in the response text during the discussion with authors (not shown by the "overall score").

---

### Decision · Program_Chairs · 2022-11-23

Accept (Poster)